biochemistry, evolution, molecular biology

biosynthetic pathways, Benzenoids, chemical ecology, Oribatid mites, Chemical defence

**Authors for correspondence:**
Adrian Brückner
e-mail: bruckner@caltech.edu
Michael Heethoff
e-mail: heethoff@bio.tu-darmstadt.de

# De novo biosynthesis of simple aromatic compounds by an arthropod (*Archegozetes longisetosus*)

Adrian Brückner[1,2], Martin Kaltenpoth[3,4] and Michael Heethoff[1]

[1]Technische Universität Darmstadt, Ecological Networks, Schnittspahnstraße 3, 64287 Darmstadt, Germany
[2]California Institute of Technology, Division of Biology and Biological Engineering, 1200 East California Boulevard, Pasadena, CA 91125, USA
[3]Evolutionary Ecology, Institute of Organismic and Molecular Evolution, Johannes Gutenberg University, Johann-Joachim-Becher-Weg 13, 55128 Mainz, Germany
[4]Department of Insect Symbiosis, Max Planck Institute for Chemical Ecology, Hans-Knöll-Strasse 8, 07745 Jena, Germany

AB, 0000-0002-9184-8562; MK, 0000-0001-9450-0345; MH, 0000-0003-3453-4871

The ability to synthesize simple aromatic compounds is well known from bacteria, fungi and plants, which all share an exclusive biosynthetic route—the shikimic acid pathway. Some of these organisms further evolved the polyketide pathway to form core benzenoids via a head-to-tail condensation of polyketide precursors. Arthropods supposedly lack the ability to synthesize aromatics and instead rely on aromatic amino acids acquired from food, or from symbiotic microorganisms. The few studies purportedly showing de novo biosynthesis via the polyketide synthase (PKS) pathway failed to exclude endosymbiotic bacteria, so their results are inconclusive. We investigated the biosynthesis of aromatic compounds in defence secretions of the oribatid mite *Archegozetes longisetosus*. Exposing the mites to a diet containing high concentrations of antibiotics removed potential microbial partners but did not affect the production of defensive benzenoids. To gain insights into benzenoid biosynthesis, we fed mites with stable-isotope labelled precursors and monitored incorporation with mass spectrometry. Glucose, malonic acid and acetate, but not phenylalanine, were incorporated into the benzenoids, further evidencing autogenous biosynthesis. Whole-transcriptome sequencing with hidden Markov model profile search of protein domain families and subsequent phylogenetic analysis revealed a putative PKS domain similar to an actinobacterial PKS, possibly indicating a horizontal gene transfer.

## 1. Introduction

Simple aromatic compounds (i.e. chemicals containing a benzene ring) are important products in chemical science and industry, but also in nature [1,2]. Overall, about 550 different simple and many more complex aromatics have been described from bacteria, fungi, plants and animals [3,4]. The unique electronic structure of the benzene ring—a delocalized π-electron system with a six-ringed carbon skeleton—provides aromatics with key structural motifs that shape interactions on both molecular and organismal levels [5,6]. In the primary metabolism of arthropods, aromatic amino acids are important building blocks of proteins, but also are used in cuticle formation, sclerotization and melanization [7,8]. Among secondary metabolites, arthropods use simple benzenoids in pheromonal communication and as defensive compounds, while the more complex polyketide aromatics also function as potent antibiotics [3,9,10].

Bacteria, fungi and plants evolved two biosynthetic routes to form benzene rings: the shikimic acid pathway [1,10] and the polyketide pathway [3]. In the latter benzene rings form via a head-to-tail condensation of poly-β-carbonyl

intermediates followed by an intramolecular condensation forming the aromatic system [11,12]. Sponges, sea urchins and some vertebrates also can synthesize simple aromatics via the polyketide pathway [3,13,14]. Among arthropods, aromatics are important for chemical interactions in insects, arachnids and myriapods [9], yet it is still unclear whether these benzenoids are acquired from diet or from endosymbiotic bacteria, or are synthesized de novo by the animals [3,10,15]. It appears that especially the complex, large ringed polyketide aromatics are mostly produced by bacteria [16–20], while simple aromatics may originate from a potential arthropod polyketide pathway [21–24]. Pankewitz & Hilker [15] reviewed available studies on polyketides and found no unequivocal evidence for de novo biosynthesis of aromatics in arthropods, since potential endosymbiont contributions were not excluded. A decade later, this still holds true [10,25,26]. Only the studies by Bestmann *et al.* [21] on aromatic trail pheromones of formicine ants and Pankewitz *et al.* [27] on anthraquinones isolated from leaf beetle eggs indicated that bacteria may not be involved in the biosynthesis of these chemicals, yet in both studies the success of antibiotic treatment was not confirmed. Also, leaf beetle anthraquinones are formed via a eukaryotic polyketide folding mode, rendering a direct bacterial involvement unlikely [28].

The oribatid mite *Archegozetes longisetosus* Aoki (figure 1*i*) produces defensive secretions containing two simple aromatics, 2-hydroxy-6-methyl-benzaldehyde (2,6-HMBD) and 3-hydroxybenzene-1,2-dicarbaldehyde (γ-acaridial), in a pair of opisthonotal exocrine glands [26]. We reared animals in a controlled, sterile environment to investigate their ability to synthesize these benzene-ringed compounds de novo. We performed stable isotope labelling experiments with different precursors under intensive antibiotic treatment to maintain symbiont-free animals and to uncover the biosynthetic pathway of these aromatics. We demonstrate that 2,6-HMBD and γ-acaridial are both synthesized de novo, probably from poly-β-carbonyl (e.g. acetyl-CoA and malonyl-CoA). Furthermore, whole-transcriptome sequencing (RNAseq) uncovered putative PKS domains similar to those of bacteria, indicating a potential horizontal gene transfer.

## 2. Results and discussion

The defensive chemicals of the oribatid mite *A. longisetosus* consist of a blend of ten compounds (figure 1*a,i*) including two terpenes (approx. 45%), six hydrocarbons (approx. 15%) and two aromatic compounds (approx. 40%) [29,30]. While the hydrocarbons probably serve as solvents, the terpenes and aromatics are bioactive compounds used as alarm pheromones and predator repellents [31].

In a first feeding experiment, we tested if elimination of potentially symbiotic gut bacteria influences the biosynthesis of 2,6-HMBD and γ-acaridial by feeding the mites with food containing 10% antibiotics (a mixture of amoxicillin, streptomycin and tetracycline; see methods below). We found that production of 2,6-HMBD and γ-acaridial was unaffected by treatments with individual antibiotics or with all three in concert (figure 1*b*; U-test: $z = -1.3$, $p = 0.19$; $n = 32$; electronic supplementary material, figure S1). The combined treatment, which was used in further incorporation experiments with labelled precursors, effectively eliminated bacteria in the mites (figure 1*c*; U-test: $z = -2.1$, $p = 0.027$; $n = 13$). These

results also were supported by fluorescence *in situ* hybridization (FISH), revealing that all detectable bacteria found on the food and in the alimentary tract (figure 1*d*) were eliminated in the antibiotic-treated mites (figure 1*e*). We detected no bacteria in the glands (figure 1*f*), and no or only single bacteria in the digestive caeca (figure 1*g*), irrespective of antibiotic treatment, further supporting the general absence of endosymbiotic bacteria. By contrast, bacteria on the outer cuticle of the mites remained mostly unaffected by antibiotic treatment and served as an internal control for FISH (electronic supplementary material, figure S2). Hence, the gland does not house bacteria involved in the production of defensive compounds or precursors, as described for defensive symbioses in other arthropods [16,18,32]. The paired digestive caeca, which are directly connected with the glandular tissue via a plasma mass [33], do not appear to be brood chambers for bacteria or yeasts (electronic supplementary material, figure S2 andS3). Also, in none of the numerous histological and ultrastructural studies on the ovarial ultrastructure, egg development, vitellogenesis and cleavage was there any indication of vertically transmitted prokaryotic or eukaryotic symbionts [34–38].

In a second experiment, the diet of *A. longisetosus* was supplemented with food containing 25% of a stable isotope-labelled precursor—[$^{13}C_6$, d$_7$] D-glucose, [$^{13}C_3$] malonic acid, sodium [$^{13}C_1$] acetate or [$^{13}C_6$] phenylalanine—and 10% antibiotics. To examine the incorporation and possible biosynthesis of 2,6-HMBD (figure 2*a*) and γ-acaridial (figure 2*b*), we compared selected fragment ions using mass spectrometry and calculated an enrichment factor based on relative intensities. Both aromatics showed consistently enriched [M+1]$^+$ to [M+8]$^+$-ion series for [$^{13}C_6$, d$_7$] D-glucose, [$^{13}C_3$] malonic acid and sodium [$^{13}C_1$] acetate. Enrichment was strongest for the most general source, the heavily labelled D-glucose, while the more specific β-carbonyl precursors showed less enrichment (electronic supplementary material, table S1) for both compounds (see figure 2*a* for 2,6-HMBD and figure 2*b* for γ-acaridial). This stepwise enrichment pattern indicates that short β-carbonyls joined to form complex molecules via multiple head-to-tail condensation reactions, which is indicative of a PKS-like mechanism [22–24]. When feeding [$^{13}C_6$] phenylalanine, we found no enrichment of the [M+6]$^+$-ion (electronic supplementary material, table S1; figure 2), which would be expected for the incorporation of the completely $^{13}$C-labelled benzene ring [39]. Instead, we again found a stepwise enrichment in the [M+1]$^+$ to [M+8]$^+$-ion series (electronic supplementary material, table S1; figure 2), indicating that only fragments of the benzene ring of [$^{13}C_6$] phenylalanine had been incorporated into 2,6-HMBD and γ-acaridial via a PKS-like reaction after the mites had catabolized the amino acids to poly-β-carbonyl units [40,41]. As we can exclude a direct influence of bacteria (antibiotics treatment, qPCR, microscopy; figure 1; electronic supplementary material figure S2), fungi (microscopy; electronic supplementary material figure S3) and protists (microscopy) [34–38], this strongly supports a de novo biosynthesis of both aromatic compounds.

Given this mass spectrometric evidence, the mite must ultimately possess the necessary enzymes to produce 2,6-HMBD and γ-acaridial via a PKS-like pathway. To explore whether *A. longisetosus* expresses the respective genes encoding for such an enzyme and to trace down its potential evolutionary origin, we performed whole-body RNAseq of adult mites,

*Proc. R. Soc. B* **287**: 20201429

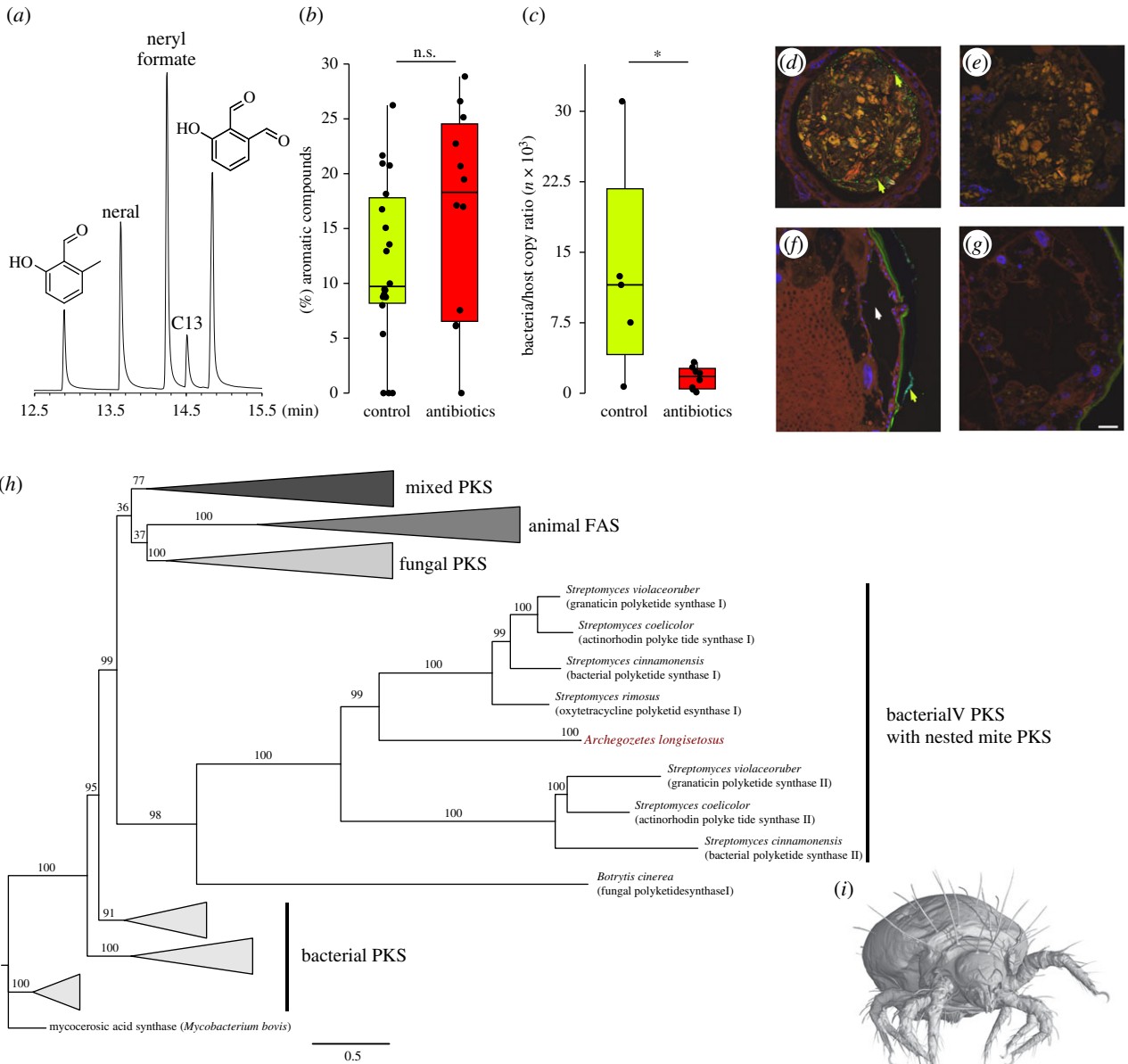

**Figure 1.** Mites (*Archegozetes longisetosus*) contain two simple aromatic compounds in their defensive secretions, and reduction of bacteria did not affect their production. (*a*) Representative GC trace of mite gland extracts; in order of retention time: 2-hydroxy-6-methyl-benzaldehyde (2,6-HMBD), neral, neryl formate, tridecane, 3-hydroxybenzene-1,2-dicarbaldehyde (γ-acaridial). Pentadec-7-ene, pentadecane, heptadeca-6,9-diene, heptadec-8-ene, heptadecane are not shown. (*b*) Supplementing a wheat grass diet (control) with a mixture of antibiotics (10% w/w; combined amoxicillin, streptomycin and tetracycline; 'antibiotics') did not affect the relative amount of aromatic compounds but resulted in significantly lower bacterial load in the mite (*c*). Fluorescence *in situ* hybridization (FISH) revealed a high bacterial prevalence in the food bolus of control group mites (*d*), while no detectable bacteria (*e*) were found in similar bolus in antibiotic-treated mites. Even in the control group, no bacteria were detected in the gland (*f*); the white arrowhead marks the centre of the gland) or in the caeca (i.e. pairwise sac-like organ that are located close to the glandular tissue; (*g*). Maximum-likelihood tree (*h*) based on an alignment of the KS-domains of fatty-acid and PKSs from animals (including those of the mite—depicted in dark red), fungi and bacteria. Bootstrap values (based on 1000 replicates) are indicated along branches and the scale bar below the tree denotes substitutions per site. The tree was rooted by the outgroup mycocerosic acid synthase from *Mycobacterium bovis*. Frontal view (*i*) of a μCT-reconstruction of *A. longisetosus*. In the FISH images, bacteria are stained in green with the general bacterial probe EUB338-Cy5; the green arrow heads mark bacterial signals detected in the food bolus (*d*) or on the cuticle (*f*). The scale bar in FISH images is 20 μm. *U*-test: n.s. = not significant, $p > 0.05$; *= significant, $p < 0.05$. (Online version in colour.)

assembled the transcriptome and predicted the longest open reading frame (ORFs) of the transcripts. Subsequent searching of the predicted protein sequences for ketoacyl-synthase (KS) domains—one module of both fatty acid (FAS) and polyketide synthases (PKSs) commonly used for phylogenetic analyses of these enzyme families [13,14]—revealed 10 candidate KS-domains in *A. longisetosus* (see Material and methods; electronic supplementary material, table S2). A phylogenetic analysis of the mite's KS-domains using a maximum-likelihood (ML) approach—including a pre-existing dataset of 139 KS-domains

[13,14] of bacterial, fungal and animal PKS as well as animal FAS—revealed that eight of the mite's KS-domains were well nested within the animal FAS-clade (see electronic supplementary material, figure S4 for full ML tree), while two KS-domains were nested with a clade of bacterial PKS, specifically PKS from the actinobacterial genus *Streptomyces* (figure 1*h*). As expected, a SPARCLE (=Subfamily Protein Architecture Labelling Engine) [42] search identified both protein sequences as beta-ketoacyl-synthases known from both FAS and PKS, yet based on their phylogenetic placement with bacterial KS-domains (figure 1*h*),

Proc. R. Soc. B **287**: 20201429

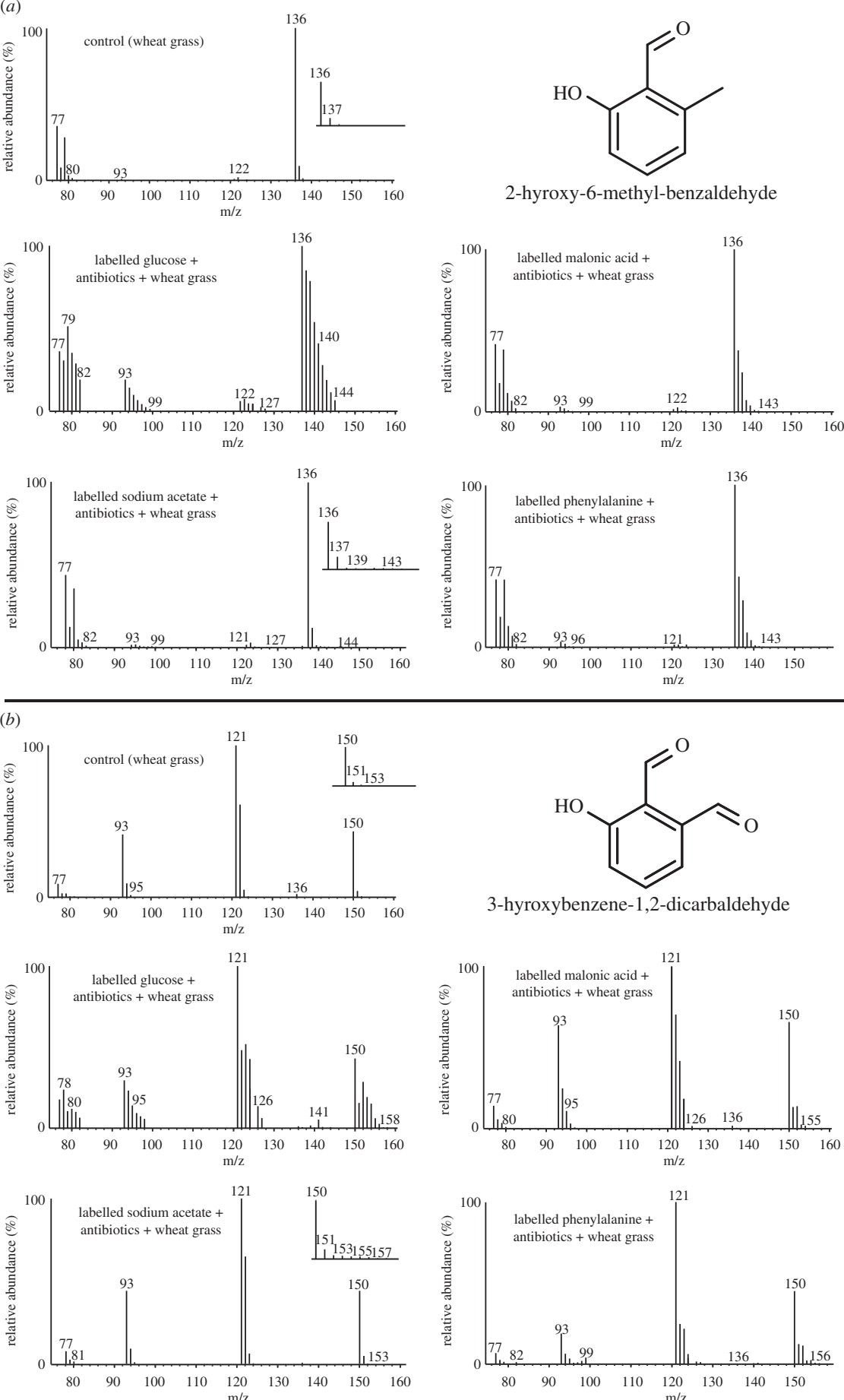

**Figure 2.** Representative mass spectra of 2-hydroxy-6-methyl-benzaldehyde (*a*) and 3-hydroxybenzene-1,2-dicarbaldehyde (*b*) extracted from defensive glands of mites fed with unlabelled wheatgrass (control), or wheatgrass infused with [13]C/d-labelled glucose, [13]C-malonic acid, [13]C-acetate and [13]C-phenylalanine recorded in single-ion mode. Inserts show the [M + 1][+]-ion series in the control and [13]C-acetate. Mites fed with wheatgrass infused with a mixture of labelled precursors and antibiotics show enriched ions.

**Figure 3.** Proposed biosynthetic scenario leading to 2-hydroxy-6-methyl-benzaldehyde (2,6-HMBD) in the oribatid mite *Archegozetes longisetosus* starting with D-glucose and phenylalanine as primary substrates. Common metabolic pathways are labelled in green, pathway-specific reactions are depicted in purple. Multiple arrows indicate multiple reaction steps. (Online version in colour.)

we refer to them as putative PKS domains. Both putative PKS domains had a GC content of approximately 37%, which is slightly higher than the usual GC content of known oribatid mite genomes (approx. 30%) [43], but much lower than in *Streptomyces* genomes (approx. 70%) [44]. Furthermore, the putative PKS domains have an amino acid composition more similar to those of other mites than to those of *Streptomyces* (electronic supplementary material, figure S5). Thus, the putative PKS domain genes are most likely integrated into the mite's genome and not due to *Streptomyces* contamination. The nested position within a *Streptomyces* clade (figure 1*h*) suggests an ancient horizontal gene transfer (HGT) of PKS-encoding genes from bacteria to *A. longisetosus*, as previously discussed for arthropods by Pankewitz & Hilker [15].

We were not able to isolate the entire biosynthetic PKS gene cluster for *A. longisetosus*, if such clusters indeed exist in animals. This might be due to the many short transcripts in the transcriptome and a lack of confirmed, contiguous gene models, but also to the diffuse genomic organization of pathway loci within the genomes of animals. Unlike in prokaryotes, where components of a biosynthetic cluster are conveniently organized as operons [45], identifying all components of biosynthetic pathways is much more challenging in animals [46]. Enzymes (or domains) are seldom clustered as tandems, but instead are scattered across the genome, and regulatory elements controlling expression are cryptic and sometimes even very distant from the open reading frame [47]. Furthermore, since there is no functional arthropod PKS-cluster for reference, it remains unknown whether PKS gene clusters of arthropods—in case they exist—are similar to those of bacteria, fungi or other animals. Our study provides evidence that HGT—a common mechanisms of acquiring new genes in mites [48] as well as other arthropods [49,50]—could explain the presence of putative PKS-domains and the mode of benzenoid synthesis via head-to-tail condensation in *A. longisetosus* (figure 3).

Since all experimental data (mass spectrometry of stable isotopes and RNAseq) point to a polyketide-like mechanism, we propose the following as the most likely biochemical pathway (figure 3): the mites harness (poly)-β-carbonyls like malonyl-CoA and/or acetyl-CoA produced via glycolysis and gluconeogenesis from sugars and amino acids, respectively, to form a C8-polyketid-like intermediate via a head-to-tail condensation. Subsequently, a ring closing cyclization via an aldol-reaction yields a second intermediate, which eventually undergoes several $-H_2O$ enolization and $-H_2O$ reduction reactions [22], inducing aromatization of the

ring and the formation of the final product 2,6-HMBD. The second aromatic compound (3-hydroxybenzene-1,2-dicarbaldehyde) may be biosynthesized from 2,6-HMBD by an enzymatic oxidation of the methyl group to the corresponding aldehyde or via a different head-to-tail condensation.

This biosynthetic scenario accords well with studies on benzenoid ant pheromones [21,23,24] and benzoquinoid defensive chemicals of harvestmen [22] that indicated a polyketide origin of these compounds, but did not exclude or specifically test for an involvement of symbiotic microorganisms [51–53]. A more closely related arachnid, the storage mite *Chortoglyphus arcuatus* Troupeau, produces an aliphatic polyketide-derived aggregation pheromone—(4R,6R,8R)-4,6,8-trimethyldecan-2-one—synthesized from one acetate and four propionates [54], further supporting the ability of mites to produce polyketides de novo, yet potential bacterial participation was not excluded, as well.

Overall, the mass spectrometric analysis of labelled precursors under controlled conditions, as well as the molecular evolutionary assessment, indicates that this oribatid mite produces small aromatic compounds using a horizontally acquired putative PKS. With more examples, we may find that ancient horizontal gene transfer had a more general role in the evolution of aromatic compound synthesis across different arthropod groups.

# 3. Material and methods

## (a) Mites

The lineage 'ran' [55] of the pantropical, parthenogenetic oribatid mite *Archegozetes longisetosus* was used in this study. Experimental cultures were established from an already existing line fed on wheat grass (*Triticum* sp.) powder from Naturya (Bath, UK) as follows: first, we collected eggs and surface-washed them with sodium hypochlorite solution (3% w/v), ethanol (70% v/v) and sterilized water for 5 s, 15 s and 30 s, respectively. Eggs were then transferred to sterile Petri dishes (diameter 45 mm) lined with 1 cm sterilized plaster of Paris. Sterile cultures were maintained in a laminar-flow closet at 28°C and 90% relative humidity. Sterilized water and 3–5 mg wheat grass or different food mixtures (see below) were provided three times each week.

## (b) Antibiotics feeding experiment

In the first experiment, we fed four different mixtures of antibiotic-laden wheat grass (10% antibiotics, w/w) and pure wheat grass as a control to different groups of 150 mites for one generation

(approx. 50 days). We prepared 10% (w/w) mixtures of sterilized wheat grass powder with amoxicillin, streptomycin and tetracycline as well as a mixture of all three antibiotics (3.3% w/w for each). One week after the adult eclosion, the defensive glands were extracted in hexane and chemically analysed (see below).

## (c) Feeding experiments with labelled precursors

For the second experiment, we used only the 10% (w/w) mixture which contained all three antibiotics. Additionally, we added 25% (w/w) stable isotope-labelled precursors. We prepared four different mixtures with [$^{13}C_6$, $d_7$] D-glucose, [$^{13}C_3$] malonic acid, sodium [$^{13}C_1$] acetate and [$^{13}C_6$] phenylalanine (all greater than 99% enrichment, Sigma-Aldrich, St Louis, USA) as well as a control with untreated wheat grass. Again, cultures were maintained for one generation and glands of adult specimens were extracted one week after eclosion using hexane (see below). Furthermore, a subset of these mites was used for FISH and qPCR experiments (see below).

## (d) Fluorescence *in situ* hybridization

For the control as well as the [$^{13}C_6$, $d_7$] D-glucose and [$^{13}C_6$] phenyl-alanine treatments (both with 10% antibiotics, see above) three entire specimens of *A. longisetosus* were fixated in 4% paraformaldehyde in PBS, and FISH was performed on semi-thin sections as described previously [56,57]. The fixated samples were dehydrated in a graded ethanol series and then embedded in cold-polymerizing resin (Technovit 8100; Heraeus Kulzer, Hanau, Germany) according to manufacturer's instructions. Sections of 7 μm thickness were obtained with a steel knife on a HM355S microtome (Leica, Germany) and mounted on microscope slides coated with poly-L-lysine (Kindler, Freiburg, Germany). FISH was done with the general eubacterial probes EUB388-Cy5 (5′-GCTGCCTCC CGTAGGAGT-3′) [58] and EUB784-Cy3 (5′-TGGACTACCAGGG TATCTAATCC-3′) [59] (two individuals each), or a combination of EUB338-Cy3 and the general yeast probe PF2-Cy5 (5′-CTCT GGCTTCACCCTATTC-3′) [60] (one individual each). Samples were incubated for 90 min at 60°C in 100 μl hybridization buffer (0.9 M NaCl, 0.02 M Tris/HCl pH 8.0, 0.01% SDS) containing 5 μl of each probe (500 nM) as well as DAPI (4′,6-diamidino-2-phenylindole) for counterstaining of host cell nuclei. Two wash steps with pre-warmed washing buffer (0.1M NaCl, 0.02M Tris/HCl pH8.0, 0.01% SDS, 5 mM EDTA), the second for 20 min at 60°C, as well as rinsing with dH$_2$O, served to remove residual probes. After drying at room temperature, slides were covered with VectaShield and inspected on an AxioImager.Z1 fluorescence microscope (Zeiss, Jena, Germany).

## (e) Quantitative PCR

To assess the effect of antibiotic treatment on absolute numbers of bacteria associated with the mites, bacterial 16S rRNA copy numbers were determined by quantitative PCR (qPCR). Since FISH experiments had revealed the presence of bacteria on the surface, the mites were surface washed in 5% (v/v) sodium dodecyl sulfate solution before bacterial quantification. For the control and the [$^{13}C_6$, $d_7$] D-glucose + 10% antibiotics treatment, DNA was extracted from eight replicates of 15–25 mites each, using the MasterPure DNA purification kit (Epicentre Technologies) according to manufacturer's instructions. As a quality control of the DNA extracts and for later standardization of bacterial titres, the DNA extracts were subjected to a qPCR with primers targeting the host 28S rRNA gene (D3A_F: 5′-GACCCGTCTTGAAA CACGGA-3′; and D3B_R: 5′-TCGGAAGGAACCAGCTACTA-3′) [61]. Subsequently, samples that showed amplification for the host 28S (five of the control replicates and all eight of the antibiotic treatment) were subjected to a qPCR with general eubacterial 16S rRNA gene primers (Univ16SRT-F: 5′-ACTCCTACGGGAGG

CAGCAGT-3′; Univ16SRT-R: 5′-TATTACCGCGGCTGCTGGC -3′) [62]. qPCRs were done on a RotorGene-Q cycler (Qiagen, Hilden, Germany) in final reaction volumes of 25 μl, including the following components: 1 μl of DNA template, 2.5 μl of each primer (10 μM), 6.5 μl of autoclaved distilled H$_2$O, and 12.5 μl of SYBR Green Mix (Qiagen, Hilden, Germany). PCR conditions included 95°C for 5 min, followed by 40 cycles of 95°C for 10 s, 70°C for 15 s, and 72°C for 10 s. A melting curve analysis was performed by increasing the temperature from 60°C to 95°C within 20 min. Standard curves were established for the host 28S and bacterial 16S assays by using $10^3$–$10^{10}$ copies of PCR product as templates. A Qubit fluorometer (Thermo Fisher Scientific) was used to measure DNA concentrations for the templates of the standard curve. The ratio between absolute copy numbers of bacterial 16S and host 28S (= bacterial/host copy ratio) was used as a standardized measure of bacterial abundance per mite sample.

## (f) Chemical analysis

Gland exudates, containing the two studied aromatics, were extracted from living mites by submersing individuals (antibiotics feeding experiment) or groups of 15 (labelling experiment) in 50 μl hexane for 3 min, which is a well-established method to obtain oil gland compounds from mites [29,30]. Crude hexane extracts (2–5 μl) were analysed with a GCMS-QP2010 Ultra gas chromatography—mass spectrometry (GCMS) system from Shimadzu (Kyōto, Japan) equipped with a ZB-5MS capillary column (0.25 mm × 30 m, 0.25 μm film thickness) from Phenomenex (Torrance, USA). Hydrogen was used a carrier gas with a flow rate of 3.00 ml min$^{-1}$, with splitless injection and a temperature ramp was set to increase from 50°C (5 min) to 210°C at a rate of 6°C min$^{-1}$, followed by 35°C min$^{-1}$ up to 320°C (for 5 min). Electron ionization mass spectra were recorded at 70 eV and characteristic fragment ions were monitored in single ion mode.

## (g) Data analysis

For the antibiotic feeding experiment, we quantified the ion abundance and calculated the relative composition of aromatics (2,6-HMBD and γ-acaridial combined) compared to ion abundance of the other compounds on an individual base. Then we compared the (%) aromatic compounds among groups or treatments with a Kruskal–Wallis test or Mann–Whitney $U$-tests, respectively. The bacterial/host copy ratio was analysed with a Mann–Whitney $U$-test as well. Statistics were performed in PAST 3.17 [63]. For stable isotope enrichment, we compared the four treatment groups with the control and calculated the enrichment factors (EF) as EF = $(r_{treatment} − r_{control})/r_{control}$, where $r_{treatment}$ is the relative abundance (% relative to M$^+$) of a respective ion in the treatment GCMS analyses and $r_{control}$ is the relative intensity (%) of the same ion in the control group.

## (h) DNA/RNA extraction, genome/RNA sequencing

For both extractions, about 200 adult mites were taken from the non-treated stock culture, starved for 24 h to avoid possible contamination from food in the gut, subsequently washed with 1% SDS for 10 s. Finally, DNA or RNA was extracted from living specimens using the 'Quick-DNA Miniprep Plus Kit' or the 'Quick-RNA MiniPrep Kit' from Zymo Research (Irvine, CA, USA) according to the manufacturer's protocols, respectively. Amounts and quality of DNA/RNA were accessed using a Qubit fluorometer and NanoDrop One (Thermo Fisher Scientific), respectively. Extracted DNA/ RNA was shipped to Omega Bioservices (Norcross, GA, USA) on dry ice for library preparation and sequencing. DNA library preparation followed the KAPA Hyper-Prep Kit protocol, RNA was used for poly-A selection, cDNA synthesis and library preparation followed the Illumina TruSeq

mRNA Stranded Kit protocol. Both libraries were 150 bp paired-end sequenced on a HighSeq4000 platform.

## (i) Genome and transcriptome assembly

Quality of reads was assessed using FastQC v. 0.11.8, Illumina adapters were trimmed with Cutadapt v. 1.18 [64]. Illumina short reads for the genome were assembled using the Platanus Genome Assembler pipeline v. 1.2.4 [65], yielding an assembly with a total length of 172.0 Mbp, an N50 = 25 196 bp and a BUSCO score [66] of C:94.4%[S:92.6%, D:1.8%], F:1.8%, M:3.8%. For the genome-guided assembly of the transcriptome, a bam-file was created from the genome fasta-file using STAR [67], while RNAseq reads were *in silico* normalized and subsequently used together with the bam-file to assemble the transcripts using Trinity v. 2.8.4 [68], yielding an assembly with a total length of 162.8 Mbp, an N50 = 2994 bp and a BUSCO score of C:96.3%[S:36.5%, D:59.8%], F:1.3%, M:2.4%. TransDecoder (v. 5.5.0) [69] was used to identify putative candidate coding regions in the transcriptome data and translate them into protein sequences.

## ( j) HMMR search, alignment and phylogenetic analysis

One way of identifying putative PKS domains is a phylogenetic analysis of the KS-domain. Therefore, we isolated potential KS-domain sequences of fatty acid and PKSs, by searching the transdecoded protein sequences with profile hidden Markov models as implemented in HMMER v3 [70] using the pfam domains 'PF16197', 'PF00109' and 'PF02801' as queries. Subsequently, protein sequences of fatty acid and PKS s previously used for phylogenetic analyses [13,14], as well as sequences from the acariform mite species *Tetranychus urticae* [48] were downloaded and the same HMMER searches were performed to extract the KS-domain sequences from the respective fasta-files. All putative KS-domain protein sequences were concatenated and sequences shorter than 300 amino acids were removed. All remaining sequences were aligned using MUSCLE [71]; ends were manually inspected and trimmed. The resulting final protein sequence alignment (electronic supplementary material, table S2; GenBank MT849380) was used to construct a maximum-likelihood (ML) phylogenetic tree for the KS domains using the iqtree pipeline [72], including automated model selection. The ML tree was constructed under the LG + R7 model using 1000 bootstrap runs and rooted by the outgroup sequence for mycocerosic acid synthase from *Mycobacterium bovis*. The similarity of amino acid frequency of the mite and *Streptomyces* KS-domains was assessed using kmer counting and principal component analysis in R [73].

Ethics. There are no legal restrictions on working with mites.

Data accessibility. The datasets supporting this article have been uploaded as part of the electronic supplementary material.

Authors' contributions. A.B. designed research, performed chemical analysis and transcriptomic work, analysed the data and took lead in drafting the manuscript; M.K. performed molecular and microscopic research, analysed the data and helped drafting the manuscript; M.H. designed research, performed chemical analysis, analysed the data and drafted the manuscript. All authors gave final approval for publication and agree to be held accountable for the work performed therein.

Competing interests. The authors declare no conflict of interest.

Funding. A.B. is a Simons Fellow of the Life Sciences Research Foundation and was previously supported by a PhD scholarship from the German National Academic Foundation. This study was supported by the German Research Foundation (DFG; HE 4593/5-1) to MH, a pilot grant of Caltech's Center for Environmental Microbial Interactions (CEMI; CEMI-19-028) to A.B. and a Consolidator Grant of the European Research Council (ERC CoG 819585 'SYMBeetle') to M.K.

Acknowledgements. We thank Roy A. Norton for improving the manuscript and Benjamin Weiss, Dagmar Klebsch and Maximilian Maschler for their technical assistance. We are further grateful to Julian Wagner who helped with bioinformatics.

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
