## [Reviewer comments · Proceedings of the Royal Society B: Biological Sciences]

Review History

RSPB-2020-1429.R0 (Original submission)

Review form: Reviewer 1

Recommendation

Accept with minor revision (please list in comments)

Scientific importance: Is the manuscript an original and important contribution to its field?

Good

General interest: Is the paper of sufficient general interest?

Good

Quality of the paper: Is the overall quality of the paper suitable?

Acceptable

Is the length of the paper justified?

Yes

Should the paper be seen by a specialist statistical reviewer?

No

Do you have any concerns about statistical analyses in this paper? If so, please specify them explicitly in your report.

No

It is a condition of publication that authors make their supporting data, code and materials available - either as supplementary material or hosted in an external repository. Please rate, if applicable, the supporting data on the following criteria.

Is it accessible?

Yes

Is it clear?

Yes

Is it adequate?

Yes

Do you have any ethical concerns with this paper?

No

Comments to the Author

The research by Brückner and collaborators is an important contribution to chemical ecology, particularly to our understanding of semiochemical biosynthesis. In this paper, to demonstrate the biosynthetic potential of mite aromatic compounds, stable isotope-labeled precursors were incorporated after administration of antibiotics to eliminate the effects of enterobacteria. In addition, the authors aimed to identify PKS gene clusters by RNAseq. The experimental method, interpretation of experimental data, and consideration of the results are appropriate. I recommend this paper to be published after the modifications and clarifications described below.

1. If no antibiotics are given, what are the gut bacteria that are mainly found in mites? The intestinal flora may be associated with PKS. If you have already analyzed the flora, you should show the data.
2. The authors used sodium acetate for stable isotope uptake experiments. On the other hand, since malonic acid is used, I think that acetic acid should be used originally.
3. Judging by the MS spectra, glucose was well metabolized by mites, but when sodium acetate and malonic acid were used, the metabolites were only as labeled as phenylalanine. Therefore, the administration of sodium acetate and malonic acid may not be the optimal precursors for this uptake experiment. It is possible that the mites eat little or that their precursors negatively affect the growth of the mites.

Review form: Reviewer 2

Recommendation

Accept with minor revision (please list in comments)

Scientific importance: Is the manuscript an original and important contribution to its field?

Excellent

General interest: Is the paper of sufficient general interest?

Excellent

Quality of the paper: Is the overall quality of the paper suitable?

Good

Is the length of the paper justified?

Yes

Should the paper be seen by a specialist statistical reviewer?

No

Do you have any concerns about statistical analyses in this paper? If so, please specify them explicitly in your report.

No

It is a condition of publication that authors make their supporting data, code and materials available - either as supplementary material or hosted in an external repository. Please rate, if applicable, the supporting data on the following criteria.

Is it accessible?

Yes

Is it clear?

No

Is it adequate?

No

Do you have any ethical concerns with this paper?

No

Comments to the Author

Brueckner et al. provide evidence for the de novo biosynthesis of aromatic defense compounds in an oribatid mite. By addition of antibiotics to the mite's diet and application of stable isotope precursors the authors show that the compounds are not of endosymbiotic origin and do not seem to be derived directly from phenylalanine. The authors speculate that the metabolites are synthesized by the polyketide biosynthetic pathway. Based on transcriptome analysis, they have identified a putative polyketide synthase that seems more closely related to bacterial polyketide synthases, which raises the possibility of an ancient horizontal gene transfer.

This is an interesting study since there has been no clear evidence yet for a de novo biosynthesis of benzenoid compounds in arthropods. Overall, the manuscript is well written and the experiments have been to the most part carefully performed and documented. Results from the FISH experiments and microscopy are quite impressive. Nevertheless, I would like to add the following comments to the authors to further improve the manuscript:

1. Gland exudates were obtained by submersing whole mites in hexane. How can the authors be sure that the extract represents specific gland exudates and not other compounds on the surface of the mite?
2. In the stable isotope labeling experiment the authors observe a pattern of stepwise enrichment in the $[M+1]^+$ to $[M+8]^+$ -ions. The authors should explain in more detail how this pattern can be explained as a consequence of polyketide biosynthesis presented in Fig. 3.
3. The authors performed a phylogenetic analysis based on ketoacyl-synthase (KS) domains. They found two mite KS domains nested within a clade of bacterial KS domains (*Streptomyces*); yet, the authors state that the two domains have an amino acid composition more similar to those of other mites? Doesn't this contradict itself? While some sequence information is provided in the

supplemental materials, it would be good to present an actual alignment of the KS domains from mites and *Streptomyces*. Also, the phylogeny is focused on the KS domains but the phylogenetic relationship based on amino acid sequences of full length proteins is not presented or discussed. I find this important if the authors claim that a horizontal gene transfer might have occurred.

4. The authors state that they were unable to isolate the entire biosynthetic PKS gene cluster. Does this mean that such a cluster exists, and if so, could more information be provided about the cluster?

5. Since a specific putative PKS has been identified, it would be informative to show expression patterns of this gene in select tissues of the mite by qRT-PCR. Is the gene associated with the gland tissue?

Minor comments:

- Figure S2: For a non-expert, it is difficult to understand what exactly the frontal section of the mite is. Could this be better presented by showing where the section occurs by using the mite model? It would help to point out more specifically where the fecal pellet is positioned. I could not see a fecal pellet in b). Is the alimentary tract empty in this case?
- Figure S3: ...detection of fungi...
- Correct acetat with acetate.
- Figure 1: It is not really clear to me why the tree is part of this figure even though the phylogenetic relationships are discussed at the end of the manuscript.

Decision letter (RSPB-2020-1429.R0)

20-Jul-2020

Dear Dr Brückner:

Your manuscript has now been peer reviewed and the reviews have been assessed by an Associate Editor. The reviewers' comments (not including confidential comments to the Editor) and the comments from the Associate Editor are included at the end of this email for your reference. As you will see, the reviewers and the Editors have raised some concerns with your manuscript and we would like to invite you to revise your manuscript to address them.

Research ethics:

Use of animals and field studies:

It is a condition of publication that you make available the data and research materials supporting the results in the article. Please see our Data Sharing Policies (<https://royalsociety.org/journals/authors/author-guidelines/#data>). Datasets should be deposited in an appropriate publicly available repository and details of the associated accession number, link or DOI to the datasets must be included in the Data Accessibility section of the article (<https://royalsociety.org/journals/ethics-policies/data-sharing-mining/>). Reference(s) to datasets should also be included in the reference list of the article with DOIs (where available).

If you wish to submit your data to Dryad (<http://datadryad.org/>) and have not already done so you can submit your data via this link [http://datadryad.org/submit?journalID=RSPB&manu=\(Document not available\)](http://datadryad.org/submit?journalID=RSPB&manu=(Document%20not%20available)), which will take you to your unique entry in the Dryad repository.

Please submit a copy of your revised paper within three weeks. If we do not hear from you within this time your manuscript will be rejected. If you are unable to meet this deadline please let us know as soon as possible, as we may be able to grant a short extension.

Best wishes,
Dr Sasha Dall
mailto: proceedingsb@royalsociety.org

Associate Editor

Comments to Author:

Your manuscript has been reviewed by two experts in your field, and their evaluation is in accord with my own: you have accomplished an exciting and novel study that is appropriate in scope and impact for this journal. However, both reviewers have raised some questions about methodology, data presentation and interpretation that will need to be addressed in a revised manuscript. I must emphasize that responding to these concerns is not optional. To the points raised by the reviewers, I will add one additional (and minor) concern - please revise Figure 3 and its legend so that readers can better understand its logic (a cornerstone of your study), including the need to identify the compounds in your alternative pathways. This journal has a reach far beyond chemical ecology or biochemistry, and the deductive approach that you have taken should be appreciated by all readers of Proc B, especially if you can help to translate the exquisite language of metabolism. I look forward to receiving your revision.

Reviewer(s)' Comments to Author:

Referee: 1

Comments to the Author(s)

The research by Brückner and collaborators is an important contribution to chemical ecology, particularly to our understanding of semiochemical biosynthesis. In this paper, to demonstrate the biosynthetic potential of mite aromatic compounds, stable isotope-labeled precursors were incorporated after administration of antibiotics to eliminate the effects of enterobacteria. In addition, the authors aimed to identify PKS gene clusters by RNAseq. The experimental method, interpretation of experimental data, and consideration of the results are appropriate. I recommend this paper to be published after the modifications and clarifications described below.

1. If no antibiotics are given, what are the gut bacteria that are mainly found in mites? The intestinal flora may be associated with PKS. If you have already analyzed the flora, you should show the data.
2. The authors used sodium acetate for stable isotope uptake experiments. On the other hand, since malonic acid is used, I think that acetic acid should be used originally.
3. Judging by the MS spectra, glucose was well metabolized by mites, but when sodium acetate and malonic acid were used, the metabolites were only as labeled as phenylalanine. Therefore, the administration of sodium acetate and malonic acid may not be the optimal precursors for this uptake experiment. It is possible that the mites eat little or that their precursors negatively affect the growth of the mites.

Referee: 2

Comments to the Author(s)

Brueckner et al. provide evidence for the de novo biosynthesis of aromatic defense compounds in an oribatid mite. By addition of antibiotics to the mite's diet and application of stable isotope

precursors the authors show that the compounds are not of endosymbiotic origin and do not seem to be derived directly from phenylalanine. The authors speculate that the metabolites are synthesized by the polyketide biosynthetic pathway. Based on transcriptome analysis, they have identified a putative polyketide synthase that seems more closely related to bacterial polyketide synthases, which raises the possibility of an ancient horizontal gene transfer.

This is an interesting study since there has been no clear evidence yet for a *de novo* biosynthesis of benzenoid compounds in arthropods. Overall, the manuscript is well written and the experiments have been to the most part carefully performed and documented. Results from the FISH experiments and microscopy are quite impressive. Nevertheless, I would like to add the following comments to the authors to further improve the manuscript:

1. Gland exudates were obtained by submersing whole mites in hexane. How can the authors be sure that the extract represents specific gland exudates and not other compounds on the surface of the mite?
2. In the stable isotope labeling experiment the authors observe a pattern of stepwise enrichment in the $[M+1]^+$ to $[M+8]^+$ -ions. The authors should explain in more detail how this pattern can be explained as a consequence of polyketide biosynthesis presented in Fig. 3.
3. The authors performed a phylogenetic analysis based on ketoacyl-synthase (KS) domains. They found two mite KS domains nested within a clade of bacterial KS domains (*Streptomyces*); yet, the authors state that the two domains have an amino acid composition more similar to those of other mites? Doesn't this contradict itself? While some sequence information is provided in the supplemental materials, it would be good to present an actual alignment of the KS domains from mites and *Streptomyces*. Also, the phylogeny is focused on the KS domains but the phylogenetic relationship based on amino acid sequences of full length proteins is not presented or discussed. I find this important if the authors claim that a horizontal gene transfer might have occurred.
4. The authors state that they were unable to isolate the entire biosynthetic PKS gene cluster. Does this mean that such a cluster exists, and if so, could more information be provided about the cluster?
5. Since a specific putative PKS has been identified, it would be informative to show expression patterns of this gene in select tissues of the mite by qRT-PCR. Is the gene associated with the gland tissue?

Minor comments:

- Figure S2: For a non-expert, it is difficult to understand what exactly the frontal section of the mite is. Could this be better presented by showing where the section occurs by using the mite model? It would help to point out more specifically where the fecal pellet is positioned. I could not see a fecal pellet in b). Is the alimentary tract empty in this case?
- Figure S3: ... detection of fungi...
- Correct acetat with acetate.
- Figure 1: It is not really clear to me why the tree is part of this figure even though the phylogenetic relationships are discussed at the end of the manuscript.

Author's Response to Decision Letter for (RSPB-2020-1429.R0)

See Appendix A.

Decision letter (RSPB-2020-1429.R1)

07-Aug-2020

Dear Dr Brückner

I am pleased to inform you that your manuscript RSPB-2020-1429.R1 entitled "*De novo* biosynthesis of simple aromatic compounds by an arthropod (*Archezogetes longisetosus*)" has been accepted for publication in Proceedings B.

The referee(s) have recommended publication, but also suggest some minor revisions to your manuscript. Therefore, I invite you to respond to the referee(s)' comments and revise your manuscript. Because the schedule for publication is very tight, it is a condition of publication that you submit the revised version of your manuscript within 7 days. If you do not think you will be able to meet this date please let us know.

Sincerely,
Dr Sasha Dall
Editor, Proceedings B
<mailto:proceedingsb@royalsociety.org>

Associate Editor:

Board Member

Comments to Author:

Thank you for your responses to the reviewers' requests and comments - your revised manuscript will constitute an exciting contribution to this journal. Please consider changing your color scheme in Figure 3, as some readers with red-green colorblind condition will not be able to distinguish your alternative colors.

Author's Response to Decision Letter for (RSPB-2020-1429.R1)

See Appendix B.

Decision letter (RSPB-2020-1429.R2)

11-Aug-2020

Dear Dr Brückner

I am pleased to inform you that your manuscript entitled "*De novo* biosynthesis of simple aromatic compounds by an arthropod (*Archezogetes longisetosus*)" has been accepted for publication in Proceedings B.

Your article has been estimated as being 9 pages long. Our Production Office will be able to confirm the exact length at proof stage.

Open Access

Paper charges

Sincerely,

Proceedings B

Appendix A

Caltech

Division of Biology and Biological Engineering

Dr. rer. nat. Adrian Brückner
1200 E. California Blvd., MC 216-76
Pasadena, CA 91125
bruckner@caltech.edu

Response letter

Dear Dr. Dall,

Please find below our response to the reviewers comments. Comments of the associated editor and reviewers are in **bold**, our responses are in *italics*. We hope that the manuscript is now ready to be published in Proceedings B.

Best from California,

Associated editor:

Please revise Figure 3 and its legend so that readers can better understand its logic (a cornerstone of your study), including the need to identify the compounds in your alternative pathways. This journal has a reach far beyond chemical ecology or biochemistry, and the deductive approach that you have taken should be appreciated by all readers of Proc B, especially if you can help to translate the exquisite language of metabolism. I look forward to receiving your revision.

—> Fig.3 was modified accordingly and we added a description of the pathway in the discussion part, so the reader can now better understand the proposed steps of the pathway.

Referee 1:

1. If no antibiotics are given, what are the gut bacteria that are mainly found in mites? The intestinal flora may be associated with PKS. If you have already analyzed the flora, you should show the data.

The composition of the gut bacterial community of oribatid mites has been previously studied Gong et al. (2018, SBB 123:155-164) and they demonstrated that there the gut communities are closely correlated with the mites' phylogeny. However, this is not linked to the chemicals that mites produces. Furthermore, Gong et al. found that Actinobacteria (the phylum that contains Streptomyces) is not a very abundant group of bacteria in mites (see their figure below).

We performed bacterial 16sRNA barcoding on a large array of mites and an unpublished studies on the influence of different food sources and nutritional mixing on the mites microbiome done by us, did not show an abundant Actinobacteria community (see red box in barplot figure). Also a unpublished genomic study by one of us (AB) utilizing long-read sequencing, reveled no significant actionabcterial contamination in the mites (see blobtools plot).

Overall, we did not get the feeling that including parts of the bacterial 16sRNA data would really add anything to the manuscript expect for a list of bacterial species. If the eitor wants such a list to be included in the supplement, we can do it.

2. The authors used sodium acetate for stable isotope uptake experiments. On the other hand, since malonic acid is used, I think that acetic acid should be used originally.

Acetic acid (whether ^{13}C -labeled or not) is a corrosive liquid and mites would certainly not survive consumption. Sodium acetate, however, is a salt, non hazarad and even commercially used as a food additive. On the other hand ^{13}C disodium malonate is not comercially available.

3. Judging by the MS spectra, glucose was well metabolized by mites, but when sodium acetate and malonic acid were used, the metabolites were only as labeled as phenylalanine. Therefore, the administration of sodium acetate and malonic acid may not be the optimal precursors for this uptake experiment. It is possible that the mites eat little or that their precursors negatively affect the growth of the mites.

As mentioned in the methods section and the results, we fed [$^{13}\text{C}_6$, d_7] D-glucose, i.e. here 13 atoms in the glucose molecule are heavier, for [$^{13}\text{C}_3$] malonic acid and sodium [^{13}C] acetate, however, only three and one atom are labeled, respectively. While differences in uptake, metabolism and feeding behavior is certainly a possibilty, the differences in heavier atoms might actually account for the MS patterns.

Referee 2:

1. Gland exudates were obtained by submersing whole mites in hexane. How can the authors be sure that the extract represents specific gland exudates and not other compounds on the surface of the mite?

*Submersing mites in hexane for a short periode of time (here 3 min) is a well-established method to only extract gland compounds. For instance the Kuwahara group (see review Kuwahara. 2004. *Advances in Insect Chemical Ecology*. pp 76–109) used this method for nearly 50 years in over about 200 studies on astigmatid mites. Similary others and we have established this method for our model organism (Sakata & Norton. 2003. *Int J Acarol* 29:345-350; Heethoff & Rasponig. 2011. *Acarologia* 51:229-236). Other surface compounds require very long extraction times, usually several days due to the cerotegument found on the cuticle of many mites (Raspotnig & Krisper. 1998. *Biosyst Ecol Ser* 14:215-243; Raspotnig et al. 2008. *Ann Zool* 58:445-452). We added parts of these information to the methods section to clarify.*

2. In the stable isotope labeling experiment the authors observe a pattern of stepwise enrichment in the [M+1]⁺ to [M+8]⁺-ions. The authors should explain in more detail how this pattern can be explained as a consequence of polyketide biosynthesis presented in Fig. 3.

We revised figure 3 according to the editors suggestion and also added some longer explanation to the main text.

3. The authors performed a phylogenetic analysis based on ketoacyl-synthase (KS) domains. They found two mite KS domains nested within a clade of bacterial KS domains (Streptomyces); yet, the authors state that the two domains have an amino acid composition more similar to those of other mites? Doesn't this contradict itself?

In a phylogenetic analysis of an alignment the KS domain clusters with the Streptomyces, which indicates that there is an evolutionary relationship (horizontal gene transfer). On the other hand, however, if there is true HGT, the transferred gene is now integrated into the mites genome and most possess "mite"-like characteristics, e.g. similar GC-%, or an amino acid composition similar to mites. If there was no similarity in GC-% or AA composition at all, it would rather indicate that the sequence is a bacterial cross contamination. Hence, recovering the mite putative PKS nested within a bacterial clade, but with mite like compositional characteristics, actually provide evidence of HGT and genomic integration and not the opposite.

While some sequence information is provided in the supplemental materials, it would be good to present an actual alignment of the KS domains from mites and Streptomyces.

As suggested by the reviewer we provide the alignment for the KS domains of the mites and Streptomyces (as sequence data) and an additional figure that we reference in the main text. Showing the alignment in the main part of the MS does not seem to add much to the overall story, hence we only added it as supplemental material.

Also, the phylogeny is focused on the KS domains but the phylogenetic relationship based on amino acid sequences of full length proteins is not presented or discussed. I find this important if the authors claim that a horizontal gene transfer might have occurred.

As we mentioned in the methods, usually only the KS domain of PKS is used to analyse phylogenetic relationships of these enzyme family as only this domain provides enough signal to make any phylogenetic claims (see Cooke et al. 2017. Cell 171:427-439; Castoe et al. 2007. Gene 392:47-58). As we also mentioned in the main text and related to the next comment of reviewer #2, we were not able to isolate the full length, as we don't know whether arthropod PKS as a clustered structure or diffused genomic loci.

4. The authors state that they were unable to isolate the entire biosynthetic PKS gene cluster. Does this mean that such a cluster exists, and if so, could more information be provided about the cluster?

After this statement we also discuss that it's unlikely, yet also unknown whether such clusters would exist in arthropods. To make this more clear we slightly modified the text

5. Since a specific putative PKS has been identified, it would be informative to show expression patterns of this gene in select tissues of the mite by qRT-PCR. Is the gene associated with the gland tissue?

While it would be great to get expression patterns for the putative PKS via qPCR, the gland itself is only ~50µm and consists only of a single cell-layer, hence tissue specific dissections seem hardly possible.

Minor comments:

Figure S2: For a non-expert, it is difficult to understand what exactly the frontal section of the mite is. Could this be better presented by showing where the section occurs by using the mite model? It would help to point out more specifically where the fecal pellet is positioned. I could not see a fecal pellet in b). Is the alimentary tract empty in this case?

-> We added an overview cartoon to make the sections easier to understand. And there is indeed no fecal pellet in (B)

Figure S3: ...detection of fungi... -> Done.

Correct acetat with acetate. -> Done.

Figure 1: It is not really clear to me why the tree is part of this figure even though the phylogenetic relationships are discussed at the end of the manuscript.

-> The phylogeny has been placed in Fig 1, to avoid another displayed item in the manuscript. For now, we'd like to leave the tree in Fig 1, but can move it/split it, if the edior wants us to do so.

Appendix B

Caltech

Division of Biology and Biological Engineering

Dr. rer. nat. Adrian Brückner
1200 E. California Blvd., MC 216-76
Pasadena, CA 91125
bruckner@caltech.edu

Response letter II

“Thank you for your responses to the reviewers' requests and comments - your revised manuscript will constitute an exciting contribution to this journal. Please consider changing your color scheme in Figure 3, as some readers with red-green colorblind condition will not be able to distinguish your alternative colors.”

The color scheme was changed to viridis colors, which should be fine to read in case of a red-green colorblind condition.

Additionally, we added the NCBI accession number that we meanwhile received to the manuscript.

Best from California,
